# Evaluating the Critical Barriers to Green Construction Technologies Adoption in China

**Yujing Wang** [1], **Dan Chong** [2,*] **and Xun Liu** [3]

1 School of Management Studies, Shanghai University of Engineering Science, Shanghai 201620, China; wangyujing_82@163.com
2 School of Management, Shanghai University, Shanghai 200444, China
3 School of Civil Engineering, Suzhou University of Science and Technology, Suzhou 215009, China; liuxun8127@usts.edu.cn
* Correspondence: chongdan@shu.edu.cn; Tel.: +86-139-1748-9887

**Abstract:** Green construction technologies (GCTs) are important drivers of sustainable development in the construction industry. Despite a wide range of GCTs being available on the Chinese construction market, they are not yet widely popular. This study aims to evaluate the critical barriers hampering large-scale GCT adoption in China. Through a literature review, 21 barriers were identified and listed in the questionnaire survey, and 225 valid responses from 21 provinces in China were collected. The Mann–Whitney U test was conducted to verify whether different stakeholder groups perceive these barriers differently. Moreover, a comparative analysis of barriers to GCT, GBT (green building technologies), and GC (green construction) adoption was conducted. Results of statistical analyses showed that the top five barriers inhibiting GCT adoption are "lack of government incentives", "extra costs associated with GCTs", "dependence on traditional construction technology", "a shortage of technological training for project staff", and "conflicts of interest among stakeholders in GCT adoption". Moreover, the top five factors preventing the adoption of GCTs differ from those of GBTs and GCs. This study not only provides valuable resources for stakeholders to better understand the critical factors preventing GCT adoption, but also could help policy makers to effectively promote GCT adoption.

**Keywords:** green construction technologies; barriers; questionnaire survey; construction industry

## 1. Introduction

The construction industry is recognized as a major contributor to social and economic development, but it has harmful impacts on the ecological environment [1–3]. Globally, the construction industry accounted for 35% of the total energy and 38% of process-related $CO_2$ emission in 2020 [4]. China has the largest construction market worldwide, and its annual construction-related $CO_2$ emissions are more than 2.1 billion tons, with 46.5% attributable to the industry overall [5]; in addition, it also generates 40% of all solid wastes [6]. However, construction is undoubtedly a national pillar of industry [7]. According to latest data published by China Statistical Bureau [8], the total output value of the construction sector increased by 6.2% to 26,394.7 billion Yuan, accounting for 25.98% of China's GDP, and more than 53.67 million employment opportunities were provided by the construction industry in 2020. Due to the detrimental impacts on the environment and pillar status of construction industry, Chinese government authorities attach great importance to sustainable construction. In 2017, the Chinese General Office of the State Council released a paper entitled "Suggestions on Promoting Sustainable and Healthy Development of Construction Industry from General Office of the State Council", which aims to provide the impetus for sustainable construction [9]. As the world's largest emitter of greenhouse gases, China set its first long-term climate goal with a 2060 carbon neutrality target.

The application of green practices is a solution to meet the challenges of climate change and environmental pollution. Therefore, the construction industry attempted to create new development paradigms by applying diverse green practices, such as green building (GB) [10,11], green construction (GC) [7,12,13], green building technologies (GBTs) [1,14], green procurement [3,15], and so on. Green construction technologies (GCTs), as one kind of GBTs, are defined as "the technologies that are applied in construction phase of a project to meet the requirements of sustainable development, GCTs can be both the green improvement of traditional construction technologies and newly developed special construction technologies". The definition of GCTs is highly related to GC and GBTs, but among them there are also differences, as shown in Table 1. The purpose of adopting GCTs, GC, and GBTs is the same, as all three are applied for resource conservation, environmental protection, and eventually realizing the sustainable development of the construction industry. Compared to GC, which are composed of technology, organization, and management, GCTs are only related to the technology factor; unlike GBTs, which are incorporated into design, construction, and operation phases of a building, GCTs are only applied in the construction phase. In summary, GCTs belong to one kind of GBTs, and are a leading factor in GCs. The scopes of GCTs, GC, and GBTs are shown in the first column in Table 1.

**Table 1.** Comparisons among GCTs, GC, and GBTs.

| Green Practice | Full Name | Purpose | Key Factor | Stages They Are Applied in |
|---|---|---|---|---|
| GC | Green Construction | Resource conservation and environmental protection; eventually realizing the sustainable development of construction industry. | Technology, organization, and management | Construction phase |
| GCTs | Green Construction Technologies | | Technology only | Construction phase |
| GBTs | Green Building Technologies | | Technology only | Design, construction, and operation phases |

Note: Table is summarized according to references [1,12,16].

With the target of facilitating the integration of innovative technologies with the construction industry, the Ministry of Housing and Urban-Rural Development of the People's Republic of China issued a document entitled, "Ten Kinds of New Technologies in Construction Industry (TNTCI)" in 2017 [17]. It promotes 11 specific GCTs, and eight of these were first introduced in the revised version of TNTCI in 2017.

Because of the sustainable benefits of GCTs, the application of GCTs played a prominent role in the sustainable development of construction industry. However, the construction industry was notorious for its slow innovation process and path dependency [18], and the adoption of GCTs is not free of challenges and obstacles [19]. The penetration of GCTs did not occur, and their contributions to resource conservation and environmental protection during construction activities are quite limited. To encourage the widespread adoption of GCTs, the barriers inhibiting adoption need to be understood first.

Previous studies primarily focused on the barriers to GBT adoption, but only limited studies were carried out on GCT barriers. For instance, Chan et al. [1] reported 20 obstacles to GBT adoption; Ghanaian. Darko et al. [20] revealed that resistance to change, a lack of knowledge and awareness, and higher cost were the top three barriers to the adoption of GCTs in the United States. Barriers hindering the adoption of specific green building technology, such as building-integrated photovoltaics [21], high-efficiency windows [22], and prefabricated technology [13,23] were also identified in many studies. The previous studies on barriers to green technology adoption primarily focused on either GBTs without separately analyzing technologies that are implemented in different phases of the building lifecycle or a specific technology that should be adopted at the design stage of a project. There are knowledge gaps regarding adoption obstacles of GCTs, which are a type of

GBT that's only implemented at the construction stage. GBT adoption is incorporated at different phases which may face different types of barriers. Contractors are the main adopters of dust control technology during the construction stage, while clients are the decision makers regarding the adoption of prefabricated building technology. Therefore, conducting a study or survey on the major barriers hampering the implementation of GCTs is worthwhile to enrich the knowledge of barriers to green practice applications in the construction industry. Furthermore, a better understanding of the factors would be helpful to develop targeted policies that overcome these barriers, which is extremely important for countries with large-scale construction operations, such as China.

Thus, this paper aims to answer three proposed questions: (1) what are the critical factors hindering GCT adoption in China; (2) whether different stakeholders' perceptions of the barriers hampering GCT adoption are different; (3) whether GCTs face unique adoption barriers compared to other green practices, such as GBTs and GC. This paper provides a thorough and comprehensive analysis of the various barriers to GCT adoption in China from a multistakeholder perspective.

## 2. Literature Review

Green practices are imperative to solve environmental issues in the construction industry. Nevertheless, the transition towards sustainable development faces enormous barriers. Targeting different kinds of green practices, such as implementation of GC, general GBTs, and special GBTs, many studies investigated the factors prohibiting their adoption.

### 2.1. Barriers to GC

A considerable amount of literature was published on the obstacles hampering GC adoption. Shi et al. [12] showed that higher costs, increased time, and limited availability of green suppliers and information are critical factors inhibiting implementation in China. Djokoto et al. [24] found that a demand shortage for green buildings, a lack of strategy to promote green construction, higher costs, low awareness of green technologies, and poor government support were important factors. An interesting finding from Zhou et al. [25] was that although respondents agreed to involve green principles in construction at a high level, the level of adoption was much lower than that of green awareness. Apart from the these barriers, long pay back periods and a reliance on traditional practice are also identified as the most significant barriers, according to Wang [26]. Lam et al. [16] argued that the factors influencing the implementation of GC specifications included: (1) green technology and techniques; (2) reliability and quality of specification; (3) leadership and responsibility; (4) stakeholder involvement, and (5) guide and benchmarking systems. Additionally, some studies focused on the managerial obstacles preventing GC [27,28]. Hwang and Ng [27] revealed that project managers face extra challenges because they spend more time on preparation with GC projects and experienced difficulties with finding suitable subcontractors.

### 2.2. Barriers to General GBT Adoption

Numerous studies explored barriers impacting general GBT adoption in many countries around the world without focusing on a particular type of GBT. A Ghanaian study identified the top three obstacles as higher costs, a lack of government incentives, and a lack of financing schemes [14,23]. They classified all barriers into five groups comprising human-related barriers, government-related barriers, knowledge- and information-related barriers, market-related barriers, and cost- and risk-related barriers. Besides higher costs, Darko et al. [10] demonstrated that resistance to change and a lack of knowledge and awareness are both the most important obstacles hindering GBT implementation in the US. Chan et al. [29] carried out a global survey and found that US respondents highlighted resistance to change as the most important barrier, while Canadians emphasized both resistance to change and conflict of interests among stakeholders as the most critical bar-

riers; Australian experts cited uncertainties with adopting new technologies to be the primary barrier.

### 2.3. Barriers to Special GBTs Adoption

Other researchers looked at obstacles preventing specific types of GCT implementation in the construction industry, such as prefabricated or offsite construction (OSC), sustainable energy technologies, green roof technology, and waste reduction technology. Furthermore, applying circular economy principles to construction is regarded as a crucial way to minimize building-related resource use, energy consumption, and $CO_2$ emissions [30].

In terms of prefabricated construction, Mao et al. [31] identified a lack of government incentives, additional costs, and reliance on traditional construction methods as the three most critical obstacles. Wu et al. [13] discovered that the top five factors affecting prefabricated construction development in China are technology lock-in, incentive policies, standardization, costs, and entrepreneurial cognition. Gan et al. [23] examined the inter-relationships among obstacles impacting OSC implementation. Wuni and Shen [32] identified 120 barriers preventing the adoption of modular integrated construction by conducting a holistic international review, and they grouped them into eight separate clusters.

Regarding sustainable energy technologies, Luthra et al. [33] examined obstacles to their adoption and categorized barriers into seven groups. Du et al. [34] showed that stakeholders' unwillingness, high initial costs, and low profitability are the major factors hampering energy-saving technology adoption. Lu et al. [21] focused on the technology of building-integrated photovoltaics and discussed how the long-term payback period, high initial costs, and low energy conversion efficiency are the main barriers to implementation in Singapore.

Concerning the application of green roof technology, Shafique et al. [35] highlighted high initial costs, high maintenance costs, and roof leakage problems as the critical barriers. Besides, high maintenance costs and a shortage of incentives and promotion from governments were regarded as the major barriers against green roof systems development for existing buildings in Hong Kong [36,37]. As for waste reduction technology, Wang et al. [38] showed that main factors influencing the willingness of design units to adopt are social, market environment, government supervision, and attitude of designers.

The literature review above suggests that many barriers to the adoption of green practices in the construction industry around the world were identified, laying the foundation for this study. However, some of the previous studies only focused on the barriers hindering the implementation of GC; similarly, some remain narrow in focus, dealing only with adoption of general GBTs or specific GBTs. As mentioned before, GCTs are different from GC and GBTs, so the conclusions of related studies are less likely to be useful for understanding GCT adoption. Therefore, a study investigating the barriers specific to GCTs is worthwhile.

### 3. Methodology

### 3.1. Data Collection

To investigate GCT adoption barriers in China, a questionnaire survey was carried out for data collection. The initial measurement scales in the questionnaire were based on the literature review above to identify the potential GCT barriers. Next, a focus group meeting and a pilot study were conducted to improve the reliability and validity of the measurement scales before distributing the questionnaire. Three sustainable construction experts were invited to join the focus group meeting to clarify concerns and whether any factors should be added, deleted, or grouped with others. A total of 23 copies were collected from the presurvey. The barrier "Senior managers pay insufficient attention to GCT adoption" in the original questionnaire was amended to "Senior managers working for contractors pay insufficient attention to GCT adoption" to ensure the respondent explicitly understood the meaning of "senior managers". The barrier "The adoption of GCTs reduces the quality of construction project" was deleted because two experts argued that a project's quality

standard is a basic requirement, and if some GCTs affect project quality, they are not useful technologies. Besides, a professor with a background in questionnaire survey methods was invited to assess the coherency of the questionnaire's framework and the accuracy of technical terms. Suggestions, such as "the definition of GCTs should be added in the introduction part", were adopted. Based on the respondents' feedback, the contents and wording of the questionnaire were adjusted and improved to a two-part final version. The first part included questions to gather basic information on the respondents, such as age, work experience, construction project experience, and positions. The second part presented the 21 identified barriers, as shown in Table 2. The respondents were then asked to give their opinions on the significance level of each barrier using a five-point Likert-type scale from 1 (strongly disagree) to 5 (strongly agree), which is the most widely used approach to scaling responses in survey research. Lastly, the questionnaire was created using the Questionnaire Star application to solicit the respondents' answers online. The questionnaire in English translation can be found as Supplementary Materials on the supplementary website.

**Table 2.** Typical barriers to GCT adoption.

| Code | Barriers | Main References |
|---|---|---|
| B1 | Complexity involved in adopting GCTs | [1,27] |
| B2 | Imperfect GC technological codes and standards | [1,39] |
| B3 | Lack of quantitative assessment tools for green performance of GCTs | [12,40] |
| B4 | Low compatibility with other construction technologies | [13] |
| B5 | Lack of related green materials and equipment | [12,20] |
| B6 | Extra costs associated with GCTs | [1,13] |
| B7 | Project delays caused by implementation of GCTs | [1,12] |
| B8 | Incremental risk and uncertainties associated with GCT adoption | [10,32] |
| B9 | Senior managers working for contractors pay insufficient attention to GCT adoption | [33,41] |
| B10 | Lack of environmental protection awareness among organization managers working for contractors | [15,39] |
| B11 | Lack of knowledge and experience related to GCT application | [1,15] |
| B12 | Lack of environmental protection awareness among technicians | [13] |
| B13 | Low labor quality | [42] |
| B14 | A shortage of technological trainings for project staff | [40] |
| B15 | Dependence on traditional construction technology | [1,13] |
| B16 | Lack of government incentives | [31,39] |
| B17 | Lack of technical assistance for adopting GCTs from government | [42] |
| B18 | Lack of demonstration projects | [1,39] |
| B19 | Lack of importance attached to GC by owner | [39,42] |
| B20 | Conflicts of interest among stakeholders in GCT adoption | [16,29] |
| B21 | Limited availability of competent subcontractors | [1,41] |

The questionnaire survey was conducted from August 2019 to January 2020. The population comprised all industry practitioners with knowledge and understanding of GCT use in China. As contractors are the final users of GCTs, they were selected as the main respondents. To make the results of the survey more objective and comprehensive, clients and consultants who have knowledge and experience with green construction were also involved.

Since not the entire contractor population is known, it is impractical to use a complete probability sampling method to collect data. Thus, a snowball sampling approach, which is a kind of nonprobability sampling technique, was adopted to establish a sample group. By applying this sampling method, members of the sample group were recruited from initial subjects to generate additional subjects via chain referral [43]. The initial respondents of the questionnaire survey were carefully selected; they must have a good understanding of green construction, and many of them were considered influential persons, such as being in a high position in their firm or working in an industry association. Finally, a total of 225 valid responses were received. As for nonprobability sampling technique, it is hard to use a formula to calculate the sample size. The sample size of the survey was similar to

or more than that of the previous studies, e.g., [1,13,21]. These respondents came from 21 provinces or municipalities in mainland China, which are listed in Table 3. As shown, there were at least four provinces or municipalities from the northern, southern, eastern, and western parts of China, thereby ensuring that the responses represent the entire population studied to a certain extent.

**Table 3.** Provinces or municipalities of China under study.

| Regions of China | Provinces or Municipalities |
| --- | --- |
| The North | Beijing, Tianjin, Jilin, Liaoning |
| The South | Hubei, Hunan, Henan, Guangdong, Hainan |
| The East | Shanghai, Jiangsu, Anhui, Shandong, Zhejiang, Fujian |
| The West | Yunnan, Guizhou, Chongqing, Sichuan, Xinjiang, Shanxi |

The respondents' demographic characteristics are summarized in Table 4. The questionnaire respondents held different positions, including enterprise managerial position (13.78%), enterprise technical position (10.22%), project managerial position (53.78%), and project technical position (22.22%). Some 45.78% of respondents have over 10 years of industrial working experience, and 54.67% have participated in more than 5 construction projects. Their long work experiences and rich project experience ensures the validity and reliability of the responses. A total of 64% of the respondents were contractors, 9.78% were clients, and 26.22% were consultants. Contractors, as the main target respondents, were selected from top 10 contractors in China in the year of 2019 (ranking based on general construction contracting revenue), which are China State Construction Co., Ltd. (ranking 1), China Railway Construction Co., Ltd. (ranking 2), and Shanghai Construction Group Co., Ltd. (ranking 4). China Communications Construction Co., Ltd., which ranked as the 3rd top contractor, is mainly devoted to the design and construct of ports, waterways, dredging, and bridges, etc. The research scope of this study is limited to the building construction industry, so these three largest contractors were selected as the main target respondents. The total general construction contracting revenue of the top 10 Chinese contractors is 3051.7 billion RMB Yuan, accounting for 67.97% of the top 80 Chinese contractors in 2019 [44]. This increased from 64.9% in the previous year, indicating that the leading enterprises occupying the vast majority of market share and construction industry concentration continue to grow. The three selected contractors' business scopes are all over the country; they have the highest technical and managerial levels and are the main adopter of GCTs.

*3.2. Data Analysis*

The data collected from the questionnaire survey was analyzed by IBM SPSS Statistics (Version 25.0) [45] To test the reliability of the five-point Likert scale, the internal consistency of the questionnaire survey was first assessed by the Cronbach's coefficient [46], which ranges from 0 to 1. The higher the value is, the higher the reliability of the measurement tool [47]. The calculated $\alpha$ value was 0.891, which was higher than 0.7, showing that the survey instrument we used was reliable at a 5% significance level.

The mean scores were calculated to measure the comparative importance among potential obstacles to GCT adoption, then importance rankings of barriers were determined by the mean values in descending order. If two or more barriers had the same mean scores, standard deviations were used to determine which was ranked higher. To determine whether respondents within a group respond in a consistent way, Kendall's coefficient (Kendall's W) of concordance was calculated. Its value range from 0 to 1, with higher values indicating more consensus within the group on the ranking of the factors. If the level of significance is less than 0.05, it is indicated that there is a consensus among respondents.

In our survey, all respondents were classified into two groups: respondents from contractors (group 1) and those from other stakeholders including clients and consultants (group 2). After criticality ranks of the individual barriers perceived by the two groups

were assigned, we investigated whether there were significant difference between the two groups via the nonparametric Manne–Whitney–U test. The H0 is that "the differences between mean values from two groups are not statistically significant", which can be rejected if the computed *p*-value is less than 0.05 at the 95% significance level.

**Table 4.** Demographic characteristics of respondents.

| Variable | Category | Frequency | Percentage (%) |
|---|---|---|---|
| Age | 21–30 years old | 83 | 36.89 |
| | 31–40 years old | 93 | 41.33 |
| | 41–50 years old | 35 | 15.56 |
| | Over 50 years old | 14 | 6.22 |
| Work experience | Less than 5 years | 65 | 28.89 |
| | 5–10 years | 57 | 25.33 |
| | 11–15 years | 44 | 19.56 |
| | More than 15 years | 59 | 26.22 |
| Project experience | Participated in 1–4 projects | 102 | 45.33 |
| | Participated in 5–8 projects | 44 | 19.56 |
| | Participated in 9–12 projects | 27 | 12 |
| | Participated in more than 12 projects | 52 | 23.11 |
| Positions | Enterprise managerial position | 31 | 13.78 |
| | Enterprise technical position | 23 | 10.22 |
| | Project managerial position | 121 | 53.78 |
| | Project technical position | 50 | 22.22 |
| Stakeholders | Contractor | 144 | 64 |
| | Client | 22 | 9.78 |
| | Consultant | 59 | 26.22 |

## 4. Results and Discussion

### 4.1. Ranking Analysis of the Complete Sample

The descriptive statistical analysis of the complete samples, including means, standard deviation (SD), and the ranking of the barriers that hamper GCTs implementation and its Kendall coefficient of concordance are illustrated in Table 5. The mean scores of the importance of the barriers range from 2.95–4.15. Barriers with mean scores lager than or equal to 3.50 were perceived as critical barriers preventing GCT application. Survey results indicate that 16 out of the 21 potential barriers are critical factors. Among them, "B16-Lack of government incentives" (mean = 4.15) was ranked as the most critical obstacle hindering GCT implementation in the Chinese construction market. "B6-Extra costs associated with GCTs" (mean = 4.01) was ranked as the second most critical obstacle, followed by "B15-Dependence of traditional construction technology" as the third (mean = 4.00). The fourth-and fifth-ranked barriers were "B14-A shortage of technological trainings for project staff" (mean = 3.93) and "B20-Conflicts of interest among stakeholders in GCTs adoption" (mean = 3.90), respectively. From the survey results, "B2-Imperfect GC technological codes and standards", "B7-Project delay caused by implementation of GCTs", "B4-Low compatibility with other construction technologies", "B12-Lack of environmental protection awareness among technicians", and "B8-Incremental risk and uncertainties associated with GCT adoption", with mean values less than 3.50, were regarded as noncritical barriers. The Kendall's coefficient for ranking the 21 barriers was 0.106, and the significance level was 0.000, which shows a reasonable agreement on the ranking of obstacles impacting GCT implementation among all respondents. The top five barriers are discussed below.

**Table 5.** Summary of survey results on barriers to GCT adoption.

| Code | Total Samples | | | Group 1 (Contractor) | | | Group 2 (Other Stakeholders) | | |
|---|---|---|---|---|---|---|---|---|---|
| | Mean | SD | Rank | Mean | SD | Rank | Mean | SD | Rank |
| B1 | 3.61 | 0.885 | 13 | 3.61 | 0.878 | 13 | 3.60 | 0.904 | 13 |
| B2 | 3.48 | 0.887 | 17 | 3.45 | 0.899 | 16 | 3.54 | 0.867 | 15 |
| B3 | 3.74 | 0.848 | 8 | 3.70 | 0.862 | 9 | 3.81 | 0.823 | 9 |
| B4 | 3.37 | 0.979 | 19 | 3.33 | 1.024 | 17 | 3.44 | 0.894 | 19 |
| B5 | 3.50 | 0.931 | 15 | 3.57 | 0.913 | 14 | 3.38 | 0.956 | 20 |
| B6 | 4.01 | 0.94 | 2 | 3.96 | 0.960 | 2 | 4.10 | 0.903 | 2 |
| B7 | 3.38 | 1.071 | 18 | 3.33 | 1.071 | 17 | 3.47 | 1.073 | 17 |
| B8 | 2.95 | 1.084 | 21 | 3.00 | 1.084 | 21 | 2.85 | 1.085 | 21 |
| B9 | 3.63 | 0.974 | 12 | 3.51 | 0.975 | 15 | 3.84 | 0.941 | 8 |
| B10 | 3.50 | 1.065 | 16 | 3.29 | 1.070 | 19 | 3.86 | 0.959 | 6 |
| B11 | 3.69 | 0.912 | 10 | 3.62 | 0.924 | 11 | 3.81 | 0.882 | 10 |
| B12 | 3.35 | 1.105 | 20 | 3.21 | 1.127 | 20 | 3.60 | 1.021 | 14 |
| B13 | 3.70 | 0.948 | 9 | 3.66 | 0.917 | 10 | 3.77 | 1.003 | 11 |
| B14 | 3.93 | 0.884 | 4 | 3.92 | 0.881 | 4 | 3.95 | 0.893 | 4 |
| B15 | 4.00 | 0.793 | 3 | 3.91 | 0.827 | 5 | 4.17 | 0.703 | 1 |
| B16 | 4.15 | 0.830 | 1 | 4.21 | 0.738 | 1 | 4.04 | 0.901 | 3 |
| B17 | 3.76 | 0.925 | 7 | 3.83 | 0.888 | 6 | 3.63 | 0.980 | 12 |
| B18 | 3.65 | 0.970 | 11 | 3.75 | 0.950 | 8 | 3.48 | 0.989 | 16 |
| B19 | 3.81 | 0.945 | 6 | 3.78 | 0.962 | 7 | 3.86 | 0.919 | 5 |
| B20 | 3.90 | 0.903 | 5 | 3.93 | 0.898 | 3 | 3.84 | 0.915 | 7 |
| B21 | 3.56 | 0.995 | 14 | 3.61 | 1.004 | 12 | 3.46 | 0.975 | 18 |
| Kendall's coefficient | 0.106 | | | 0.112 | | | 0.141 | | |
| Level of significance | 0.000 | | | 0.000 | | | 0.000 | | |

"Lack of government incentives" (B16) was ranked as the most critical barrier hindering the GTC adoption. This finding is similar to previous studies concerning the adoption of OSC in China [31], the implementation of GBTs in Ghana [1], and the application of energy technology in China [34], where lack of government incentives was deemed the most prominent barrier or the second most important barrier. Currently in China, although a few policy incentives encouraging GCT adoption exist, they are far from sufficient. There is a perception that government, acting as a "motor", should play a more active role to speed up GCT adoption by making effective incentive policies. Implementing GCTs can bring environmental and social benefits, but additional costs caused by GCTs means profit losses to construction stakeholders. If incentive policies cannot offset the additional costs, accompanied by lack of knowledge of GCTs and their benefits, developers and contractors may hesitate to adopt GCTs in their construction projects. To accelerate the application of GCTs, government departments need to provide adequate incentive programs, including both financial incentives (e.g., subsidies) and nonfinancial incentives (e.g., more opportunities to be a demonstration project), especially in the early stage of the adoption of GCTs.

"Extra costs associated with GCTs" (B6) was ranked second among all the 21 barriers. This finding is similar to the numerous previous studies [1,13,14,21,39] where the higher cost was identified as one of top three barriers that hamper green practice adoption. The cost of GCTs is much higher than traditional construction technologies [48]. For example, if saving water technologies are adopted during construction process, extra costs must be paid for water-saving apparatus, reclaimed water treatment system, and collection of untraditional sources of water, such as that from foundation pit dewatering. In construction industry, cost is always the greatest concern for each participant when deciding whether to adopt new technologies and new norms [12]. The comprehensive benefits consisting of environmental, social, and economic benefits produced by GCT adoption are not only for the clients and contractors, but also for each stakeholder, including residents around the project. Nevertheless, clients and contractors have to pay for all the additional cost involved in GCT applications, which results in a loss of profit. Therefore, the willingness to adopt

GCTs is low. As the incentive policies are implemented, and knowledge and awareness of GCT benefits are improved, the cost obstacles may be overcome to some degree.

The barrier "Dependence of traditional construction technology" (B15) occupied the third position. This finding coincides with the previous research with regard to the barriers impacting OSC in China conducted by Mao et al. [31], where dependence of traditional construction technology was also identified as the third most important barrier. However, the finding is not aligned with the previous study concerning the factors hindering green building adoption in Vietnam [39], where reluctance to adopt changes was ranked low. GCTs belong to one kind of green innovation in construction industry, which was notorious for its slow innovation process and path dependency [18]. Productivity, quality, and product functionality of the construction industry were relatively poor in contrast with that of other industries [49]. Therefore, traditional technologies will keep their superior counterparts from taking off, which is known as the "lock-in" phenomenon [13]. Contractors are not familiar with how to use GCTs, and this is accompanied by uncertainties and risks; besides, other barriers discussed in this study such as lack of government incentives, high costs, and a lack of demonstration projects, therefore they may be willing to choose traditional construction technology rather than GCTs.

Another important barrier was "a shortage of technological trainings for project staff" (B14) (ranked fourth), resulting in lack of knowledge relating GCT application, which was also deemed a barrier to GCTs. However, lack of training was not considered as one of the major barriers impacting the adoption of green practice adoption in many previous studies [12,13,34,50], and was ranked low in the studies on the obstacles preventing green building technology implementation in the US [2] and in Ghana [1]. According to this finding, it can be asserted that the number of contractors who are skilled in GCTs is limited. Without sufficient training for new technologies, project staff have to either learn how to use them themselves or apply them incorrectly. This consumes a lot of money and time, which decreases contractors' willingness to adopt GTCs. If project staff can gain more training before applying GCTs so they understand what to do when they encounter technical difficulties, be it from the government or their organization, the application of GCTs will accelerate in China.

"Conflicts of interests among stakeholders in GCTs adoption" (B20) occupied the fifth position. This is in contrast with the previous survey conducted by Chan et al. [1], where conflicts among stakeholders was ranked as one of the last three obstacles hampering GBT application. Meanwhile, it is worth noting that this barrier was not perceived as a critical factor hindering green technologies application in construction industry, such as energy-saving technologies [34], prefabricated construction [13] and extensive green roof systems [37]. Based on this finding, "Conflicts of interests among stakeholders in GCTs adoption" is a critical barrier to the application of GCTs, which should not be overlooked. Many problems in construction industry resulted from interest conflicts among stakeholders of construction projects, and barriers to GCT implementation are no exception. For example, the purposes of establishing corporate image and improving the possibility of being a demonstration project drive the contractor to apply GCTs, but the client pays more attention to the objectives of project quality, costs, and the schedule; thus, they do not give sufficient support or cooperate with the contractor. Such conflicts of interest may prevent GCT application. It could be crucial for GCT penetration that government policies concerning GCTs take the various stakeholders' interests into consideration.

### 4.2. Comparitive Analysis among Stakeholders

As discussed above, "Conflicts of interests among stakeholders in GCTs adoption" is a critical barrier to GCT application, so different stakeholders with different interests may hold divergent opinions on GCT application. Because the samples from clients were relatively small, consultants are entrusted by clients, and they can represent their interests. Therefore, responses from clients and consultants were grouped together in this study. Thus, all respondents were divided into two groups: contractors (group 1) and other

stakeholders consisting of clients and consultants (group 2). The views of the two groups on what prevents GCT implementation in Chinese construction industry are illustrated in Table 5.

As for the top five barriers, although their specific values and rankings were different, four of them identified by the two groups were the same: "Lack of government incentives" (B16), "Extra costs associated with GCTs" (B6), "Dependence of traditional construction technology" (B15), and "A shortage of technological trainings for project staff" (B14). This result is consistent with the analysis result of the whole sample. The one left of the top five barriers in group 1 was "Conflicts of interests among stakeholders in GCTs adoption" (B20), and in group 2 it was "Lack of importance attached to GC by owner" (B19). Contractors and other stakeholders (clients and consultants) hold almost the same opinions on the top five critical barriers hampering GCT application, and the results within the total sample were credible and reliable.

The Mann–Whitney U test was further employed to find whether there is any significant difference between the two groups' evaluations on the importance of each barrier. The test results are demonstrated in Table 6. The four barriers, which are "Dependence of traditional construction technology" (B15), "Lack of environmental protection awareness among organization managers working for contractors" (B10), "Lack of environmental protection awareness among technicians" (B12), and "Senior managers working for contractors pay insufficient attention on GCTs adoption" (B9), have significant differences between the two groups at the 5% significance level. They are all related to contractors' awareness, attitudes, and behaviors. In addition, it is interesting to note that respondents from group 1 perceived the four barriers less important than group 2. Particularly, the differences between the ranks of the barrier B10 given by the two groups were quite high: group 1 ranked B10 19th, with a low average score of 3.29, while group 2 ranked it 6th, with an average score of 3.86. The mean scores of B10 and B12 were less than 3.30 within group 1, while they were higher than 3.6 within group 2. This indicated that contractors did not believe that their environmental protection awareness was poor, and as a result, the factors B10 and B12 were not ranked by them as the critical factors preventing GCT adoption. For other stakeholders', B10 and B12 were the main barriers. A possible explanation is that during the promotion of sustainable construction by government in recent years, contractors' awareness and emphasis on environmental protection improved gradually, whereas other stakeholders lack sufficient communication with contractors on environmental protection issues.

**Table 6.** Mann–Whitney U test on the barriers to GBTs adoption.

| Test Statistics | B1 | B2 | B3 | B4 | B5 | B6 | B7 | B8 | B9 | B10 | B11 |
|---|---|---|---|---|---|---|---|---|---|---|---|
| **Mann–Whitney U** | 5783.500 | 5467.000 | 5309.000 | 5489.000 | 5283.500 | 5346.000 | 5398.500 | 5400.000 | 4571.500 | 3941.000 | 5152.000 |
| **Z** | −0.110 | −0.825 | −1.204 | −0.766 | −1.235 | −1.121 | −0.960 | −0.959 | −2.853 | −4.239 | −1.569 |
| **Asymp. Sig.** | 0.913 | 0.409 | 0.229 | 0.443 | 0.217 | 0.262 | 0.337 | 0.338 | 0.004 * | 0.000 * | 0.117 |

| Test statistics | B12 | B13 | B14 | B15 | B16 | B17 | B18 | B19 | B20 | B21 | |
|---|---|---|---|---|---|---|---|---|---|---|---|
| **Mann–Whitney U** | 4660.000 | 5413.000 | 5629.000 | 4823.000 | 5276.000 | 5280.000 | 5029.500 | 5499.000 | 5517.500 | 5297.500 | |
| **Z** | −2.605 | −0.951 | −0.467 | −2.410 | −1.281 | −1.252 | −1.808 | −0.752 | −0.714 | −1.195 | |
| **Asymp. Sig.** | 0.009 * | 0.341 | 0.641 | 0.016 * | 0.200 | 0.211 | 0.071 | 0.452 | 0.475 | 0.232 | |

* significant, *p*-value < 0.05.

### 4.3. Comparison Analysis with Other Green Practices

To provide valuable insights into developing targeted policies for promoting the application of GCTs, the top five barriers to GCT adoption in China were compared with that of GBT adoption in Ghana, as identified by Chan et al. [1] (study A), and that of GC in China, according to Shi et al. [12] (study B). The reasons these two previous studies were chosen as comparison targets are that GBTs and GC are highly related to GCTs, and although the latter study was conducted in Ghana, it is as a developing country similar to China. The rankings of the top five GCT adoption barriers in these two related studies are

shown in Table 7. The barriers that were not identified as the barriers to the green practice adoption in the selected studies are marked with the symbol "-".

**Table 7.** Rankings of top five GCT adoption barriers in highly related studies.

| Top Five Barriers to GCTs Adoption | Ranking in this Study | Ranking in Study A | Ranking in Study B |
|---|---|---|---|
| Lack of government incentives | 1 | 2 | - |
| Extra costs associated with GCTs (or GBTs, GC) | 2 | 1 | 1 |
| Dependence on traditional construction technology | 3 | 17 | - |
| A shortage of technological training | 4 | 10 | - |
| Conflicts of interest among stakeholders in GCT adoption | 5 | 24 | - |

Note: Study A: study on barriers to green building technology (GBT) adoption [1]; Study B: study on barriers to green construction (GC) [12].

The results in Table 7 present that the barrier "Extra costs associated with green practice adoption" (B6) is the only barrier found in the top five positions of all three studies. This indicates that extra costs involved in green practice adoption is not only the top barrier preventing the GCTs adoption, but also the most critical barrier prohibiting GBT and GC adoption in developing countries. As declared by Shi et al. [12], cost is always the biggest concern for each participant when deciding whether to adopt new technologies and new norms in the construction industry. Reducing the cost of green practice adoption by way of innovation and applying the method of Value Engineering can be greatly helpful to the promotion of green practice application. In addition, "lack of government incentives" was ranked as one of the top five barriers to the adoption of both GCTs in our study and GBTs in study A, but not regarded as a barrier to GC in study B.

"Dependence of traditional construction technology", "A shortage of technological trainings", and "Conflicts of interest among stakeholders in GCTs adoption" were not in the top five barriers to adoption of GBTs and GC in studies A and B. Although these three were identified as barriers to GBTs, their ranks are very different from those of the barriers to GCTs. Taking the barrier "Conflicts of interest among stakeholders in GCTs adoption" for example, it was ranked fifth in our study, while it ranked very low (24th) in study A. These three barriers were not mentioned in the context of GC application in study B.

The results of comparative analysis prove that the most important barriers to GCT adoption are quite different from those of GBTs and GC application in developing countries, which further demonstrates the necessity and importance of identifying and understanding the most prominent obstacles hindering the adoption of specific objects of GCTs. A possible explanation is that although GCTs, GBTs, and GC are highly correlated, their characteristics, scopes, applying stage, and adoption decision-makers are different. For example, the decision on whether to adopt GCTs is made in the construction stage, mainly by the contractor, while the decision on whether to adopt GBTs is made in the design stage by the developer. As mentioned above, compared with GC, GCTs mainly relate to the technology factor of GC, exclusive of the works of organization and coordination during the process of green construction.

## 5. Conclusions

This study identified the major barriers that hamper green construction technologies (GCTs) adoption in the Chinese construction industry. Through a literature review, 21 potential barriers were identified and then a questionnaire survey was conducted, with 225 valid questionnaires collected from 21 provinces in China. Statistical analysis method was further employed to provide a clear and deep understanding of the critical obstacles to GCTs. The main conclusions as follows:

(1) 16 out of the 21 potential barriers were critical barriers to implementation of GCTs. Among them, "Lack of government incentives", "Extra costs associated with GCTs", "Dependence of traditional construction technology", "A shortage of technological trainings

for project staff", and "Conflicts of interest among stakeholders in GCTs adoption" were recognized as the top five factors inhibiting GCT application.

(2) Contractors (group 1) and other stakeholders (group 2) perceived four barriers to GCT adoption significantly differently: "Dependence of traditional construction technology", "Lack of environmental protection awareness among organization managers working for contractors", "Lack of environmental protection awareness among technicians", and "Senior managers working for contractors pay insufficient attention on GCTs adoption". Moreover, group 2 assigned more importance to these four barriers than group 1. The results indicate that there is a need to take the perspectives of different stakeholders into account when making policies to mitigate the barriers to GCT adoption.

(3) A comparative analysis among barriers to GCT, GBT, and GC adoption showed that "Extra costs associated with green practice adoption" is the only common barrier among top five barriers to all of them in construction markets in developing countries. Except for the cost factor, the top five barriers inhibiting the adoption of GCTs largely differed from that of GBT and GC implementation. Although there are many similarities among GCTs, GBTs and GC, targeted policies and measures are needed to promote GCT applications.

(4) The application of green practices, which are solutions to environmental pollution issues and energy consumption in the construction industry, is facing enormous barriers. Previous studies focused on the barriers hindering the implementation of GC, general GBTs, or specific GBTs. GCTs are different from GC and GBTs, so the conclusions of previous research have limitations for the GCT adoption.

The findings of this research contribute to filling the gap in the knowledge of barriers to green practice adoption, more specifically in the context of GCTs. What's more, it's also useful for governments to establish effective and suitable policies towards overcoming the obstacles hampering the promotion of GCTs in the construction market by providing valuable information and references. The findings provide references for other countries that plan to accelerate GCTs on construction market.

However, limitations also exist in this study. Firstly, the current study only focused on the factors prohibiting GCT application. Policies and strategies for accelerating GCT application will be further investigated in a future study. Secondly, the comparative analysis was conducted based on the perceptions of contractor group and other stakeholder group, which only included clients and consultants. The respective viewpoints of other stakeholders toward the barriers to GCT adoption and whether there are any differences among their perceptions may be an interesting research direction in the future.

**Supplementary Materials:** The following are available online at https://www.mdpi.com/article/10.3390/su13126510/s1, Investigation on the Critical Barriers to Green Construction Technologies Adoption in China.

**Author Contributions:** Conceptualization, Y.W. and D.C.; methodology, Y.W.; software, X.L.; validation, Y.W., D.C. and X.L.; formal analysis, Y.W.; investigation, Y.W. and D.C.; resources, X.L.; data curation, X.L.; writing—original draft preparation, Y.W.; writing—review and editing, D.C.; visualization, X.L.; supervision, Y.W.; project administration, D.C.; funding acquisition, D.C. All authors have read and agreed to the published version of the manuscript.

**Funding:** This research was funded by Science Foundation of Ministry of Education of China (16YJC630013) and National Science Foundation of China (71901139).

**Informed Consent Statement:** Informed consent was obtained from all subjects involved in the study.

**Data Availability Statement:** The data of this study is available from the authors upon request.

**Acknowledgments:** The authors would like to thank generous help provided by the China Railway Co., Ltd., China State Construction Co., Ltd., and Shanghai Construction Co., Ltd., etc., during the data collection.

**Conflicts of Interest:** The authors declare no conflict of interest.

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
