# Peer review of "Evaluating the Critical Barriers to Green Construction Technologies Adoption in China"

_sustainability, doi:10.3390/su13126510_

Round 1

Reviewer 1 Report

Dear authors,

I reviewed your manuscript entitled ´ Evaluating the critical barriers to green construction technologies adoption in China´ where you identified barriers for the implementation of green construction technology in practice. The topic of the manuscript is very pertinent and respond to the actual stage of research. The paper is well written, structured and fluid. However, have I have some suggestion hoping to improve the paper before it can be published.

Line 30-34 you presented the burdens of construction industry in china. In this was you put limits of your paper by guiding the readers and showing them that the paper is mostly valid for the Chinese context. But I believe that your findings are also important and valid for other countries. So, I would suggest to start with the introduction with the harmful of constriction industry in the world. I recommend to focus on the data published in the international energy agency.

Table 1 and figure 1 can be merged together. You can show turn vertical the figure 1 and add it in the table in the place of column (Green practice). In this way you show the link between GCTs, GC and GBTs and also the differences between them.

Line 148-150 you are indicating the GCT such are prefabricated or off-site construction, sustainable energy technologies, etc… One of the most important GCT where a lot of nowadays Scientifics are focusing is the circular economy. You can find more information in the European long roadmaps where the circularity is amongst 7th most powerful pathway. But in term barriers this solution is facing problem in the allocation of benefits (doi.org/10.1016/j.scs.2020.102322). I believe that adding this in your manuscript will enrich the manuscript and broaden the utility in other countries.

Line 215. Please add the reference for the SPSS 25.0 software.

I would recommend to add the discussion section in the paper. Compare your findings with those of other studies. Show how your paper extent and improve the topic. If the discussion is short than it can be also merged with the conclusion.

Conclusion is very clear and supported by the findings of the paper.

For transparency reason I would recommend to add in the supplementary material the questionnaire. 

Reviewer 2 Report

The manuscript aims to evaluate the critical barriers to the adoption of green construction technologies in China. It is well written but there are several concerns that the authors need to address. These concerns need to be addressed to clarify the questions regarding the validity of research data. These are listed below.

Line 183 to 235: 3.1 Data Collection

-What is the study population?

-What type of sampling is used for the study? What is the formula used to determine the size of the sample?

-How could the 225 valid responses be used to represent the entire population studied?

-Is there a pilot study conducted to validate the questions contained in the questionnaire survey? If this has been done, please provide a detailed description of the pilot study and its findings. If this has not been done, please provide a valid justification for why the pilot study was not conducted.

-Line 195: Justify the suitability of the Likert scale to analyze the views of respondents on the level of significance of each barrier. Why have other types of scales such as Guttman, Mokken, and Proximity not been considered in the study?

-Line 200-201: Please specify the 21 provinces or municipalities of mainland China under study.

-Line 203: Please indicate the top 10 contractors in China who participated in the study.

-Why should key respondents be selected from among the top 10 contractors in China? Do you not think that the results could be limited to respondents' perceptions from the top 10 contractors only and not of the top 50 or 100 contractors in China?  So how valid is the number of the respondents to represent the entire population under study?
